# Study on Static Characteristics of Aerostatic Bearing Based on Porous SiC Ceramic Membranes

**DOI:** 10.3390/membranes12090898

**Published:** 2022-09-17

**Authors:** Xin Xiao, Jianzhou Du, Yu Zhang, Jingyi Yan, Yunping Li, Kongjun Zhu, Luming Wang

**Affiliations:** 1School of Materials Science and Engineering, Yancheng Institute of Technology, Yancheng 224051, China; 2Jiangsu Dongpu Fine Ceramic Technology Co. Ltd., Lianyungang 222000, China; 3State Key Laboratory of Mechanics and Control of Mechanical Structures, Nanjing University of Aeronautics and Astronautics, Nanjing 210016, China

**Keywords:** aerostatic bearing, porous SiC ceramic membrane, static characteristics, permeability

## Abstract

The porous aerostatic bearing is a new supporting structure that is widely used in precision and ultraprecision engineering and the aerospace and other fields. The aerostatic bearing has a good bearing capacity and static stiffness. In this work, the numerical and experimental research on the static characteristics of an aerostatic bearing based on a porous SiC ceramic membrane is presented. The porous ceramic membrane prepared by reactive sintering, with a porosity of 25.8% and a pore size of 20.55 μm, was used as the restrictor to fabricate the aerostatic bearing. It was found that the ceramics have good permeability, and the permeability coefficient reached 2.78 × 10^−13^ m^2^ using permeability-test experiments. The effects of the gas-supply pressure and permeability coefficient on the static characteristics of the aerostatic bearing based on porous ceramics were analyzed using Fluent simulation calculation. When the gas-supply pressure was 0.5 MPa and the gas-film thickness was 6 μm, the static stiffness of the aerostatic bearing reached a maximum of 20.9 N/μm, while the bearing capacity was 632.5 N. The numerical results of the static characteristics of the aerostatic bearing are highly consistent with the experimental results, which verifies the accuracy of the Fluent simulation, and provides convenience for studying the static characteristics of aerostatic bearings.

## 1. Introduction

An aerostatic bearing, which is a sliding bearing with air as the lubricating medium, uses pressured air film to support and outward the load. Typically, aerostatic bearings possess the advantages of small friction and wear, high motion accuracy, a long service life, no pollution, etc., and they have been widely used in various precision measuring instruments and in space technology and other fields [1,2,3,4,5]. According to the type of restrictor, aerostatic bearings can be divided into several categories, such as the annular orifice restrictor [6], slot restrictor [7,8], and porous restrictor [9,10]. Compared with the annular orifice restrictor and slot restrictor, which possess lower bearing capacities, stiffnesses, and stabilities, porous materials have countless tiny interconnected pores on the surface and inside. Using porous materials as the throttle of aerostatic bearings can achieve uniform pressure distribution and improve the bearing capacity and stability [11,12,13].

The permeability coefficient of porous media, which is related to the porosity, pore size, and pore structure of porous materials, largely affects the bearing capacity of aerostatic bearings [11,14,15]. Xu et al. [16] studied the factors that affect the permeability of porous SiC ceramics, and the results indicated that the permeability increases with increases in the SiC particle size, powder content, and particle size of the pore-forming agent, but with the decreasing of the firing temperature. Liu et al. [17] studied the effect of the raw SiC particle size on the permeability and pore size of prepared porous SiC ceramics. With the increasing grit designations of the SiC particles from 2000# to 120#, the permeability of the porous ceramics increased from 6.65 × 10^−15^ m^2^ to 4.85 × 10^−13^ m^2^, and the pore size increased from 1.0 μm to 12.2 μm. The application of the material in aerostatic bearings is not complete. Li et al. [18] prepared porous alumina ceramics with a pore size of 2 μm and studied their permeability. With the porosity increase from 34.9% to 54.9%, the permeability of the porous ceramics increased from 1.41 × 10^−15^ m^2^ to 9.6 × 10^−15^ m^2^.

In addition, more people have analyzed the static characteristics of aerostatic bearings based on the permeability of porous materials. Ishibashi et al. [4] illustrated the static characteristics of a downsized aerostatic circular thrust bearing with a single feed hole by solving the Navier–Stokes equation using CFD, and considering the inertia force of the airflow. Li et al. [19] proposed a novel aerostatic bearing with back-flow channels to increase the load capacity, stiffness, and stability of the aerostatic bearing. Zhao et al. [9] studied the static characteristics of ultraprecision aerostatic bearings with porous alumina ceramics as the restrictor, in which the open porosity and permeability of the porous alumina were 25% and 3.2 × 10^−15^ m^2^, respectively. When the film thickness was 7 μm, the bearing capacity reached 274.5 N. Duan et al. [20] prepared a multi-microporous stainless-steel plate with a porosity of 25%, an average pore size of less than 10 μm, and a permeability of 8.43 × 10^−11^ m^2^ for the gas bearing, and they studied its static performance. The bearing capacity was positively correlated with the gas-supply pressure and diameter of the throttle, and negatively correlated with the film thickness. 

In this study, a porous SiC ceramic membrane prepared by reactive sintering was used as the restrictor to fabricate an aerostatic bearing, and its microstructure and performance were analyzed. The bearing capacity and static stiffness of the porous aerostatic bearing were numerically solved by Fluent. The effects of the gas-supply pressure, permeability, and gas-film thickness on the bearing capacity and static stiffness of the aerostatic bearing were analyzed, and they were then verified by experiments.

## 2. Preparation and Characterization of Aerostatic Bearing

### 2.1. Structural Design and Preparation of Aerostatic Bearing Based on Porous SiC Ceramics

Generally speaking, the structure of an aerostatic bearing is mainly divided into two parts: the bearing body and restrictor. For the design of the bearing body, according to the design drawing shown in Figure 1a, aluminum alloy material was selected to be processed by the lathe into the real object shown in Figure 1b. In the process of preparing the aerostatic bearing, the most significant factor is the preparation of the restrictor. For the restrictor, porous SiC ceramic was used as the restrictor of the aerostatic bearing. 

The porous SiC ceramic membranes were fabricated by reactive sintering. The raw materials were α-SiC powders (purity: 98%, d_50_ = 10 μm), 4 wt.% of carbon powder (d_50_ = 8 μm) as the pore-making agent, 6 wt.% of carbon black (purity > 99%, d = 3–5 μm), 5 wt.% of carboxymethyl cellulose (CMC) as the binder, and 10 wt.% of dibutyl phthalate (DBP) as the dispersing agent, which were mixed in a blinder mixer for 4 h to form ceramic slurries. The ceramic slurries were refined by vacuum, and they were then aged in a sealed container at a temperature of 30 °C and a humidity of 45% for 2 days. After that, the slurries were pressed into a homemade cylindrical mold with a forming pressure of 75 MPa. After drying the pressed embryo in an oven at 80 °C, the cured SiC ceramics were carbonized in a muffle furnace at 800 °C for 4 h under argon gas protection to obtain a SiC ceramic embryo. The Si particles (d_50_ = 5 μm) were placed at the bottom of porous preforms, which were then contained in a boron nitride (BN)-coated graphite crucible. The crucible was placed in the muffle furnace for melting and infiltration sintering. This sintering process was performed at a vacuum of less than 0.1 Pa, a temperature of 1450 °C, and for a holding time of 2 h, and the preforms were then allowed to cool to room temperature. Finally, the sintered porous SiC ceramic was processed into a wafer with a diameter of 43 mm and a thickness of 5 mm. Figure 1c shows the macroscopic morphology of the prepared porous SiC ceramic membrane. Compared with traditional ceramics, the porous ceramics have dense micropores on the surface, and the pores are through-holes.

After that, the processed bearing body was bonded together with the prepared porous SiC ceramic to form an aerostatic bearing based on the porous SiC ceramic shown in Figure 1d. Figure 1e shows the aeration of the porous aerostatic bearing in water under a certain gas-supply pressure. It is worth noting that bubbles flowed evenly from the ceramic, except at the edges. Thus, the porous SiC ceramic prepared in the study had great potential in the aerostatic bearing.

### 2.2. Pore Characteristics and Mechanical Properties

The density-measurement principle of porous ceramics is as follows: the upward buoyancy of an object immersed in a liquid is equal to the gravity of the liquid displaced by the object. Deionized water was used as the immersion medium, and the density and porosity of the porous ceramics were measured by an electronic densitometer, which is called the Archimedes drainage method [20,21]. First, the gravity of the sample was weighed in the air, which is called the dry weight (*m*_1_). Then, the sample was immersed in deionized water to measure its gravity in water, which is called the liquid weight (*m*_2_). Finally, the sample was removed from the deionized water and the moisture was wiped off the surface before weighing the gravity in the air, which is called the wet weight (*m*_3_). 

The density (*ρ*) of the porous SiC ceramic was calculated as 1.866 g/cm^3^ by Equation (1), and the porosity (*P*) of the porous SiC ceramic could then be calculated as 25.8% by Equation (2):(1)$$\rho =\frac{m_{1}}{m_{3}-m_{2}}$$
(2)$$P=\left(\frac{m_{3}-m_{1}}{m_{3}-m_{2}}\right)\times 100\%$$
where *ρ* is the density of the porous SiC ceramic membrane (g/cm^3^); *m*_1_, *m*_2_, and *m*_3_ are the gravities of the porous ceramic membrane in different states (dry weight, liquid weight, wet weight (g), respectively); *P* is the porosity of the porous SiC ceramic membrane (%).

The mercury method is a common method to measure the pore size and pore size distribution of porous materials [22]. The principle is that mercury is pressed into porous materials by external pressure, and the pore size is determined by calculating the pressure required to fill a certain pore. According to the physical capillary phenomenon, if the liquid does not infiltrate the porous material (*θ* > 90°), then the presence of surface tension prevents the liquid from immersing [23,24]. Mercury is a noninfiltrating and nonreactive liquid for porous ceramics, and it is immersed into pores by pressurization to overcome the surface tension. The pore size of the porous material is determined by calculating the pressure filled with a certain pore. The pore size of porous SiC ceramic can be calculated by Equation (3) [25]:(3)$$D=2r=-\frac{4\sigma \cos\theta }{p}$$
where *D* is the pore size of the porous material (μm); σ is the surface tension of the impregnated liquid (N/m); *θ* is the wetting angle of the impregnated liquid to the tested material (°); *p* is the external pressure (Pa). When the mercury surface tension is *σ* = 0.48 N/m and the wetting angle is *θ* = 140°, Equation (3) becomes $p\times r=7.5\times 10^{8}$, which is the most commonly used mercury-injection-test equation, where *r* is the radius of the pore size (μm).

The pore size distribution of the porous SiC ceramic membrane was measured by an automatic mercury porosimeter (Type 60-GT, Quantachrome Co., LTD., Boynton Beach, FL, USA) using the mercury method. Figure 2a shows the relationship between the differential of the pore volume to the logarithm of the pore diameter and pore size. The differential varies significantly from 8.2 μm to 34.4 μm, and it first increases and then decreases, reaching a maximum of 0.7231 cc/g at the pore size of 20 μm, which represents that the pore size of the porous ceramic is 20 μm. As shown in Figure 2b, the surface area of the porous SiC decreased with the increase in the pore size. The maximum surface area was 0.2236 m^2^/g at the pore size of 0.02 μm, and when the pore size was 20 μm, the surface area reached 0.0102 m^2^/g. 

The fracture surface microstructure of the fired samples was observed by scanning electron microscopy (Type. SEM3200, Chinainstru and Quantumtech Co., Ltd., Hefei, China). Figure 2c,d shows the microstructure of the porous SiC ceramic membrane. In the enlarged structure, a stable structure with obvious interconnected pores between the SiC particles can be found. These interconnected pores are mainly generated by the burning of carbon powders and the stacking of SiC particles. Some smaller SiC particles are distributed between the original SiC particles, which is due to the new SiC generated by reactive sintering. The average pore size of the porous SiC ceramic membrane was measured by way of Nano Measure software (Ver. 1.2, Fudan University, Shanghai, China), and it was found to be 20.55 μm, which is basically consistent with that measured by the mercury method. In Ref. [26], Calderon et al. proposed an effective method to produce sawdust preforms, and the reactive infiltration of the carbon preforms had sufficient carbonization and permeable strength. The high-temperature treatment of bioCSw modifies the carbon structure, which changes the reactivity of the substrate with Si, and higher reactivity leads to the formation of fine and uniform SiC microstructures. 

In order to test the mechanical properties of the porous SiC ceramics, the three-point bending method was adopted in the electronic universal testing machine (Type. RG-4010, Shenzhen Regal Co., LTD., Shenzhen, China), with a span length of 24 mm and a loading rate of 5 mm/min, to calculate the flexural strengths of the samples, which reached 256 MPa. Meanwhile, the Vickers hardness values of the samples were determined by the indentation method with a universal microhardness tester (HV-1000Z, Shanghai Jvjing Precision Instrument Manufacturing Co. Ltd., Shanghai, China), and they were up to 1593 Hv. 

### 2.3. Permeability of Porous SiC Ceramic Membrane

The bearing capacity of an aerostatic bearing is largely determined by the permeability coefficient of the porous media [27]. In order to measure the permeability coefficient of the porous SiC ceramic membrane, a measuring device for the permeability was designed and built, the device diagram of which is shown in Figure 3. It is mainly composed of an air compressor (Type OTS-550, Taizhou Outstanding Industry and Trade Co. Ltd., Taizhou, China), a gas-drying device (Type OA-GLQ, Qingdao Yibo Purification Equipment Co. Ltd., Qingdao, China), a pressure-regulating valve with a pressure gauge (Type IR1020-01, Shenglong Pneumatic Components Co. Ltd., Ningbo, China), a flowmeter (Type MF5708, Nanjing SENLOD Measurement and Control Equipment Co. Ltd., Nanjing, China), a power supply (Type.UTP3305, UNIT Corp., Tulsa, OK, USA), and an aerostatic bearing. The processed specimens of the aerostatic bearing based on the porous SiC ceramic are shown in addition. The pressure-regulating valve is used to control the gas-supply pressure by the air compressor, and the gas is dried through the gas-drying device. The different gas flows through porous materials are measured by a flowmeter under different gas-supply pressures, and the pressure difference before and after passing through the porous materials is recorded. The permeability coefficient is calculated by Equation (4), through Darcy’s law [28]:(4)$$\varphi =\frac{Q\mu H}{\Delta pA}$$
where *H* is the thickness of the porous material (m); *A* is the cross-sectional area of the porous material (m^2^); Δ*p* is the pressure difference before and after the gas passes through the porous material (Pa); *Q* is the volume flow rate of the gas (m^3^/s); *μ* is the aerodynamic viscosity (N·s/m^2^); *φ* is the permeability coefficient (m^2^).

According to the permeability measurement experiment, the experimental data for the volume-flow-rate measurement of the porous SiC ceramics, with gas-pressure drops of 0.02 MPa, 0.03 MPa, 0.04 MPa, and 0.05 MPa, are shown in Table 1. In combination with Equation (4), the permeability coefficients of porous materials under different pressure drops can be calculated, and the permeability of porous SiC ceramics can be finally obtained by calculating their average values. The permeability of the porous SiC ceramics was 2.78 × 10^−13^ m^2^.

## 3. Simulation of Static Characteristics of Aerostatic Bearing

### 3.1. Establishment of Calculation Model

Aerostatic bearings are used in aerospace and ultraprecision machine tools, and so it is very important to study their static characteristics, which are characterized by the bearing capacity, static stiffness, and so on. In this work, the theoretical analysis of a porous aerostatic bearing under a steady state was carried out. The schematic structure of the porous aerostatic bearing is shown in Figure 4.

Before the theoretical analysis of the porous air bearing, some necessary assumptions must be made in order to establish a physical model that can reflect the actual working conditions and facilitate the theoretical analysis. The gas flow in porous material is mainly viscous flow, ignoring the influence of inertia flow and obeying Darcy’s law. The lubricating gas is isothermal, and the influences of the viscosity and density of the gas are not considered. It is an ideal gas, and it satisfies the ideal gas equation. On the critical surface of the porous material and bearing clearance, the pressure inside the porous material is equal to the pressure of the gas on the surface of the bearing clearance.

In the theory of fluid mechanics, the moving gas in bearings satisfies the mass, momentum, and energy conservation equations of compressible gas [29]. The continuity equation is as follows [11]:(5)$$\frac{\partial \rho }{\partial t}+\frac{1}{r}\frac{\partial \left(\rho rv_{r}\right)}{\partial r}+\frac{1}{r}\frac{\partial \left(\rho v_{\theta }\right)}{\partial \theta }+\frac{\partial \left(\rho v_{z}\right)}{\partial z}=0$$

The movement of gas obeys Darcy’s law in the porous restrictor, and it satisfies the Reynolds equation of gas lubrication under isothermal conditions in the gas-film gap. The relationships between the flow velocity and pressure gradient in each direction of the porous restrictor can be calculated by Equation (6), while the relationships in the bearing clearance can be calculated by Equation (7):(6)$$\left\{ \begin{array}{l} v_{r}=-\frac{\varphi }{\mu }\frac{\partial p\prime }{\partial r}\\ v_{\theta }=-\frac{\varphi }{\mu r}\frac{\partial p\prime }{\partial \theta }\\ v_{z}=-\frac{\varphi }{\mu }\frac{\partial p\prime }{\partial z} \end{array}\right.$$
(7)$$\left\{ \begin{array}{l} \frac{\partial^{2}v\prime_{r}}{\partial z^{2}}=\frac{1}{\mu }\frac{\partial p}{\partial r}\\ \frac{\partial^{2}v\prime_{\theta }}{\partial z^{2}}=\frac{1}{\mu }\frac{1}{r}\frac{\partial p}{\partial \theta }\\ \frac{\partial p}{\partial z}=0 \end{array}\right.$$
where *p′* is the internal pressure of the porous material; *p* is the internal pressure of the gas film; *ρ* is the density of the gas; *r*, *θ*, and *z* are the coordinates in the cylindrical coordinate system; *t* is the time; *v_r_*, *v_θ_*, and *v_z_* are the velocity components of the gas in the porous medium in the *r*, *θ,* and *z* directions, respectively; *v′_r_*, *v′_θ_*, and *v′_z_* are the velocity components of the gas in the gas-film gap; *μ* is the aerodynamic viscosity; *φ* is the permeability coefficient.

Darcy’s law has been widely used in the study of fluid flows in porous materials. Despite its simplicity, it has proven to be very useful. In Refs. [24,30,31], Narciso et al. considered that Darcy’s law could be obtained analytically from the Navier–Stokes equation, assuming saturation and laminar flow, and ignoring the influence of gravity. The Reynolds number (Re) could be obtained from the equation, followed as Re=ρvD/η, where *ρ* is the density of the fluid; *D* is the pore size, which can be assumed to be about 1/3 of the average size of the particles; *v* is the fluid velocity; *η* is the dynamic fluid viscosity. The Reynolds number calculated is far less than 2300, which means that the laminar flow assumption is indeed correct. In addition, the expression of the capillary number as $Ca=\mu v/\sigma_{LV}$ is also mentioned, where *μ* is the viscosity, *v* is the triple-line velocity, and *σ_LV_* is the liquid–vapor surface energies.

The gas-pressure-distribution equations of porous aerostatic bearings in porous materials and the gas-film gap underneath are as follows:(8)$$\frac{1}{r}\frac{\partial }{\partial r}\left(r\frac{\partial p\prime^{2}}{\partial r}\right)+\frac{1}{r^{2}}\frac{\partial^{2}p\prime^{2}}{\partial \theta^{2}}+\frac{\partial^{2}p\prime^{2}}{\partial z^{2}}=0$$
(9)$$\frac{1}{r}\frac{\partial }{\partial r}\left[rh^{3}\left(1+\zeta \right)\frac{\partial p^{2}}{\partial r}\right]+\frac{1}{r^{2}}\frac{\partial }{\partial \theta }\left[h^{3}\left(1+\zeta \right)\frac{\partial p^{2}}{\partial \theta }\right]=12\varphi {\left(\frac{\partial p\prime^{2}}{\partial z}\right)}_{z=0}$$
where *h* is the film thickness, and $\zeta =\frac{3\left(\varphi^{1/2}h/\alpha \right)}{h\left(h+\varphi^{1/2}/\alpha \right)}$ is the slip coefficient.

The boundary conditions satisfied by gas in bearing motion are as follows:
(1)Gas inlet boundary: 0 ≤ *θ* ≤ 2π, 0 ≤ *r* ≤ *R*, *z* = −*H*, *p*′ = *p**_s_*;(2)Continuous boundary: 0 ≤ *θ* ≤ 2π, 0 ≤ *r* ≤ *R*, *z* = 0, *p*′ = *p*;(3)Closed boundary: 0 ≤ *θ* ≤ 2π, *r* = *R*, −*H* ≤ *z* ≤ 0, ∂*p*′/∂r = 0;(4)Gas outlet boundary: 0 ≤ *θ* ≤ 2π, *r* = *R*, 0 ≤ *z* ≤ *h*, *p*′ = *p**_a_*.


### 3.2. Establishment of Geometric Model and Grid Meshing

According to the detail drawing in Figure 5a, and the parameters of the aerostatic bearing in Table 2, the geometric model was established (Figure 5b). The main structure of the aerostatic bearing includes the bearing body and porous restrictor. The type of restrictor used in this paper is a porous SiC ceramic restrictor. The principle of a porous aerostatic bearing is that the external compressed air enters the high-pressure gas cavity, and then passes through the porous restrictor, forming a gas-film gap between the restrictor and support surface, thus providing a certain bearing capacity and stiffness for aerostatic bearings [2,32]. As the Fluent software calculation is only for the fluid domain, the fluid model of the aerostatic bearing is presented in Figure 5c. It is composed of a high-pressure gas cavity, porous region, and gas-film clearance. The meshing of the entire fluid domain is shown in Figure 5d. Because the pressure of the high-pressure gas cavity is determined by the gas-supply pressure and the pressure changes little, the region is not the key calculation area of the model. The grid is relatively thick. The porous region and gas-film clearance are the critical computational areas, which require a fine grid, and especially for the gas-film clearance. The focus of the mesh division is to divide the mesh in the direction of the gas-film clearance. The thickness of the gas-film clearance is smaller than that of other regions, and the mesh number in the thickness is set as 10. When the control volume is discretized, the gas-film clearance and porous region mainly use hexahedral grid elements, and the final total number of the control volume elements is 128040.

The entire model is set as a fluid region. The inlet surface of the high-pressure gas cavity is the pressure inlet, and the circumferential surface of the gas-film clearance is the pressure outlet. The connection surfaces of the porous region with the high-pressure gas cavity and gas-film clearance are set to the jump boundary. In Fluent, the gas is assumed to be an ideal gas, and the flow in the porous region and gas-film clearance is assumed to be a laminar flow, while the effect of the wall roughness on the gas flow is ignored. The pressure inlet, environmental temperature, and pressure outlet are set to 0.5 MPa, 298 K, and 0 MPa, respectively. The two most significant parameters that define a porous medium are the viscous resistance and internal resistance. Here, the viscous resistance was set to 3.60 × 10^12^ m^−2^, but the internal resistance was not considered. 

### 3.3. Simulation Analysis of Aerostatic Bearing

The residual convergence curve and flow state of the gas in the whole fluid domain of the aerostatic bearing are shown in Figure 6, when the gas-supply pressure, gas-film thickness, and permeability coefficient were 0.5 MPa, 10 μm, and 2.78 × 10^−13^ m^2^, respectively. For the simulation analysis and calculation under this state, the convergence curve of the residual is shown in Figure 6a. After 416 steps of iterations, the residual curve was close to the flat lines, which is when it is considered that the calculation has convergence. Figure 6b,c shows the pressure distributions on the load surface. The pressure on the load surface decreased gradually from the center to the periphery, and it decreased to normal pressure at the pressure outlet. Figure 6d shows the pressure distribution in the porous media. When the gas moved in the porous media, the pressure distribution was uniform, and the pressure drop was small. The flow velocity of the gas in the gas-film gap is shown in Figure 6e. The velocity of the gas in the center of the gas film was the smallest, and it increased gradually along the radial direction. The maximum gas velocity at the circumferential edge of the gas film was 360 m/s. In Ref. [20], the mathematical model of a porous stainless-steel plate was established in Fluent based on permeability data and fluid mechanics theory. The distributions of the pressure and gas velocity on the bearing surface were simulated, which proved the consistency of the results.

### 3.4. Static-Characteristic Analysis of Aerostatic Bearing

Under the permeability of 2.78 × 10^−13^ m^2^, the effects of the static characteristics on the aerostatic bearing with different gas-supply pressures were studied (Figure 7). The relationship between the bearing capacity and air-film thickness of the aerostatic bearing with gas-supply pressures of 0.3 MPa, 0.4MPa, and 0.5 MPa is shown in Figure 7a. It can be observed that the bearing capacity decreases with the increase in the air-film thickness under the same gas-supply pressure, and the decrease is slower. Under the condition of a constant air-film thickness, with the increase in the gas-supply pressure, the bearing capacity is greater. The gas-film stiffness is the change rate of the bearing capacity, and its accuracy is related to the value density of the bearing capacity. Figure 7b shows the relationship between the static stiffness and air-film thickness under different gas-supply pressures. With the increase in the gas-supply pressure, the change trend of the bearing capacity of the aerostatic bearing was more obvious, and the static stiffness also increased accordingly. When the gas-supply pressure was constant, the static stiffness increased first, and then decreased with the increase in the gas-film thickness, and a maximum static stiffness appeared with a gas-film thickness ranging from 4 μm to 8 μm. The results showed that the static characteristics of the aerostatic bearing can be improved by increasing the gas-supply pressure on the premise of ensuring good gas continuity.

Figure 8 shows the variation curves of the bearing capacity and static stiffness with the gas-film thickness under different permeabilities of 5.0 × 10^−13^ m^2^, 2.5 × 10^−13^ m^2^, 1.0 × 10^−13^ m^2^, and 0.5 × 10^−13^ m^2^ at a gas-supply pressure of 0.5 MPa. It can be seen from the figure that the permeability of the porous media had a significant impact on the static characteristics of the aerostatic bearing. When the gas-film thickness was 4 μm, the bearing capacities of the aerostatic bearing with different permeability coefficients were 703.93 N, 687.455 N, 639.89 N, and 588.86 N, showing an obvious linear downward trend. The bearing capacity of the aerostatic bearing was negatively correlated with the gas-film thickness under the constant permeability coefficient. With the increase in the gas-film thickness, the static stiffness first increased, and then decreased. The gas-film thickness corresponding to the optimum static stiffness of the aerostatic bearing moved to the right with the increase in the permeability coefficient.

## 4. Experimental Research on Static Characteristics of Aerostatic Bearing

### 4.1. Construction and Test of Static-Characteristic-Measurement System for Aerostatic Bearing

In order to study the static characteristics of the porous aerostatic bearing, a bearing-capacity-measurement system of the aerostatic bearing was built in this study. The concrete physical figure of the experimental measurement system is shown in Figure 9. The whole measuring system consists of a power supply (Type UTP3305, UNIT Corp.), air compressor (Type OTS-550, Taizhou Outstanding Industry and Trade Co. Ltd), gas-drying device (Type OA-GLQ, Qingdao Yibo Purification Equipment Co. Ltd), pressure-regulating valve with a pressure gauge (Type IR1020-01, Shenglong Pneumatic Components Co. Ltd.), displacement transducer (Type IL-S025, Keyence Corp., Osaka, Japan), data display instrument (Type IL-1000, Keyence Corp., Osaka, Japan), compression-testing machine (Type ZQ-32, Dongwan Zhiqu Precision Instruments Co. Ltd., Dongwan, China), support plate, and aerostatic bearing [33]. The porous SiC ceramic membrane with a permeability of 2.78 × 10^−13^ m^2^ was selected as the restrictor to study the static characteristics of the aerostatic bearing.

Before the experiment, one side of the aerostatic bearing with the porous restrictor was placed on the support plate, and the other side was contacted with the compression-testing machine. During the whole experiment, the air compressor and gas-drying device provided a high-pressure source of dry gas, and then the pressure-regulating valve with the pressure gauge was used to adjust and read the appropriate pressure. The compression-testing machine provided a certain amount of pressure, which was the bearing capacity of the aerostatic bearing, and the displacement transducer was used to measure the floating height of the aerostatic bearing, which was the thickness of the gas film. According to the recorded data, the relationship between the bearing capacity and air-film thickness was plotted.

### 4.2. Experimental Verification Results of Static Characteristics of Aerostatic Bearing

The aerostatic bearing prepared with porous SiC ceramic as the restrictor, the permeability coefficient of which is 2.78 × 10^−13^ m^2^, was taken as the experimental object. For a gas-supply pressure from 0.3 MPa to 0.5 MPa, the bearing capacity measured by the experimental device for measuring the static characteristics of the aerostatic bearing shown in Figure 9 are shown in Figure 10a. The experimental results indicate that the gas-supply pressure and gas-film thickness are important factors that affect the static characteristics of aerostatic bearings. The bearing capacity increases with the increase in the gas-supply pressure, and it decreases with the increase in the gas-film thickness.

A comparison of the theoretical calculation and experimental results of the bearing capacity of the aerostatic bearing based on porous SiC ceramic at a gas-supply pressure of 0.3 MPa and a permeability of 2.78 × 10^−13^ m^2^ is shown in Figure 10b. The theoretical results are basically consistent with the experimental results, which indicates that the theoretical calculation is highly reliable. Combining the theoretical calculation and the experiment verifies the accuracy of the conclusions, which has guiding significance for the research and preparation of aerostatic bearings.

## 5. Conclusions

In short, an aerostatic bearing based on a porous SiC ceramic membrane was designed and prepared in this work. The porous SiC ceramic membrane prepared by reactive sintering, which was used as the restrictor of the aerostatic bearing, had a uniform pore distribution, porosity of 25.8%, pore size of 20.55 μm, and permeability of 2.78 × 10^−13^ m^2^. The porous aerostatic bearing has good bearing capacity and static stiffness. The mathematical model of the aerostatic bearing was established by Fluent software to analyze the effects of the gas-supply pressure, permeability coefficient, and gas-film thickness on the bearing capacity and static stiffness of the aerostatic bearing. The simulation results indicated that the bearing capacity of the aerostatic bearing increased with the increases in the gas-supply pressure and permeability, and it decreased with the increase in the gas-film thickness. The static stiffness was positively correlated with the gas-supply pressure, and it was negatively correlated with the permeability. The experimental results verify the accuracy of the simulation, and they provide a basis for the fabrication of porous aerostatic bearings and research on the static characteristics.

## Figures and Tables

**Figure 1 membranes-12-00898-f001:**
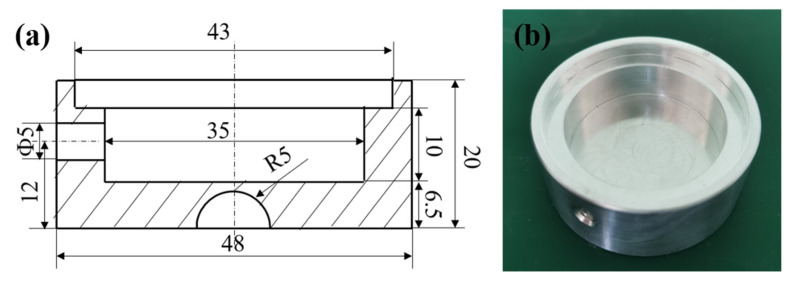
(**a**) Design drawing of bearing body; (**b**) real object of bearing body; (**c**) macroscopic morphology of porous SiC ceramic; (**d**) physical photograph of aerostatic bearing; (**e**) aeration of porous aerostatic bearing in water.

**Figure 2 membranes-12-00898-f002:**
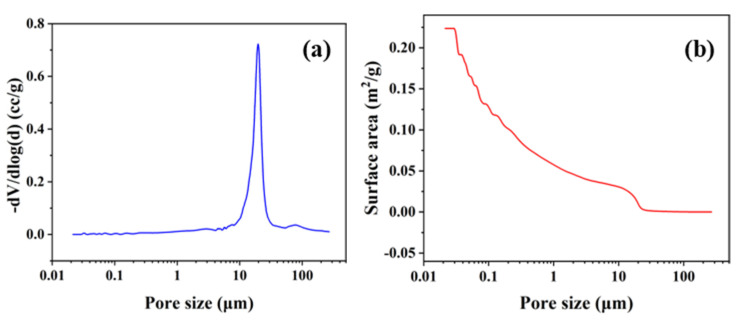
(**a**) Pore size distribution; (**b**) surface area; (**c**,**d**) fracture surface microstructure of porous SiC ceramic membranes.

**Figure 3 membranes-12-00898-f003:**
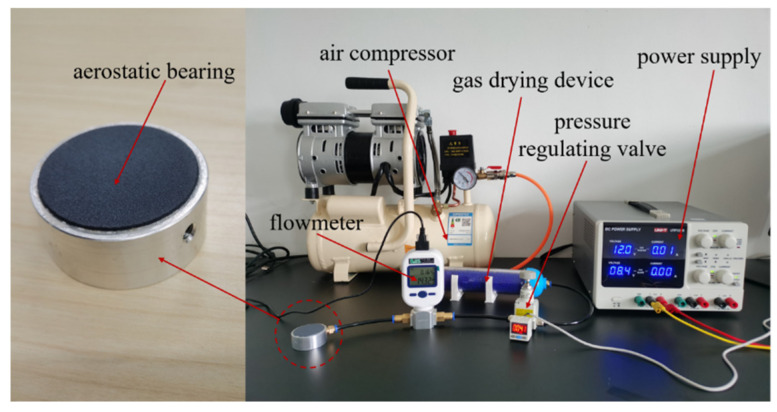
Permeability-coefficient measuring device system.

**Figure 4 membranes-12-00898-f004:**
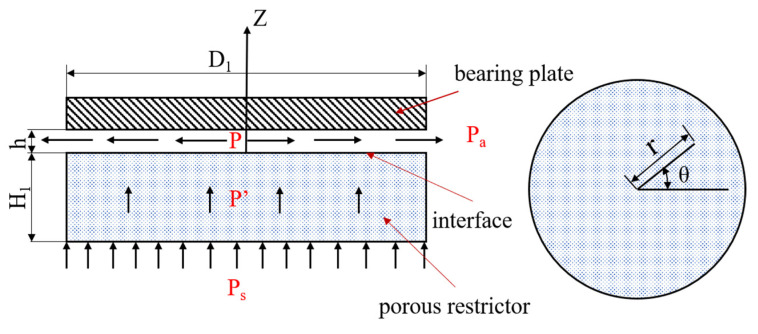
The schematic structure of the aerostatic bearing.

**Figure 5 membranes-12-00898-f005:**
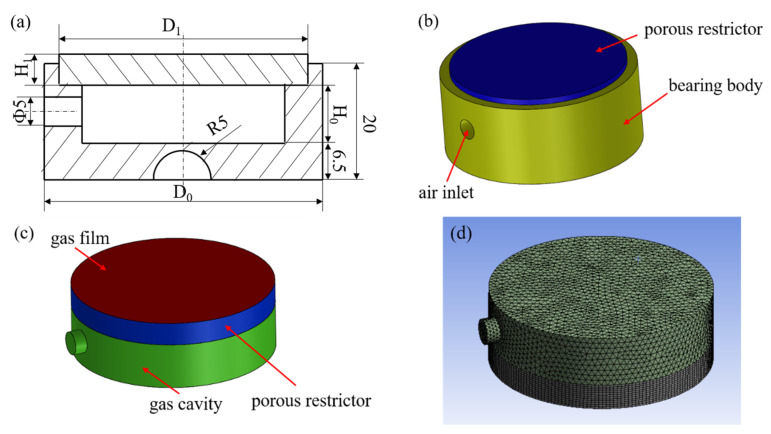
Detail drawing and simulation model of aerostatic bearing: (**a**) detail drawing; (**b**) geometric model; (**c**) fluid model; (**d**) meshing.

**Figure 6 membranes-12-00898-f006:**
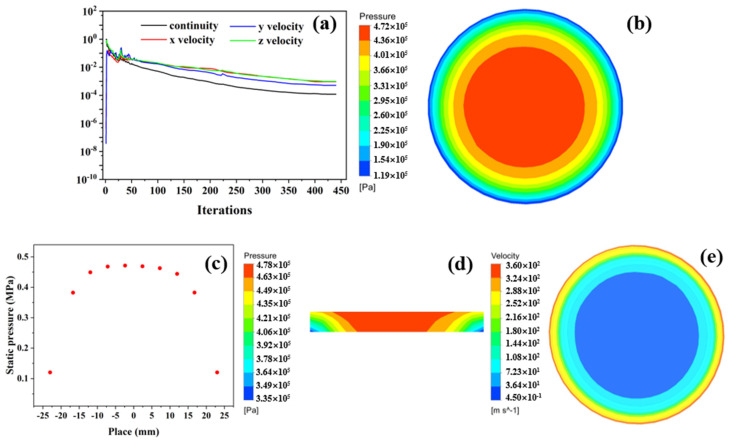
The residual convergence curve and flowing gas state under the conditions of Ps = 0.5 MPa, h = 10 μm, and φ = 2.78 × 10^−13^ m^2^: (**a**) residual convergence curve; (**b**) pressure contours of load surface; (**c**) two-dimensional distribution; (**d**) pressure distribution on profile of porous media; (**e**) gas-flow velocity of bearing clearance.

**Figure 7 membranes-12-00898-f007:**
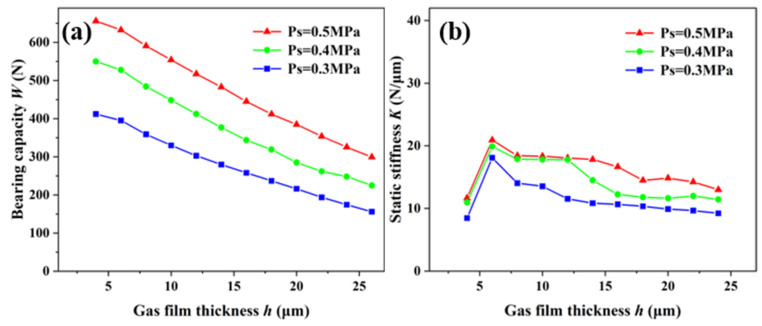
Static characteristics of aerostatic bearing under different gas-supply pressures: (**a**) bearing capacity; (**b**) static stiffness.

**Figure 8 membranes-12-00898-f008:**
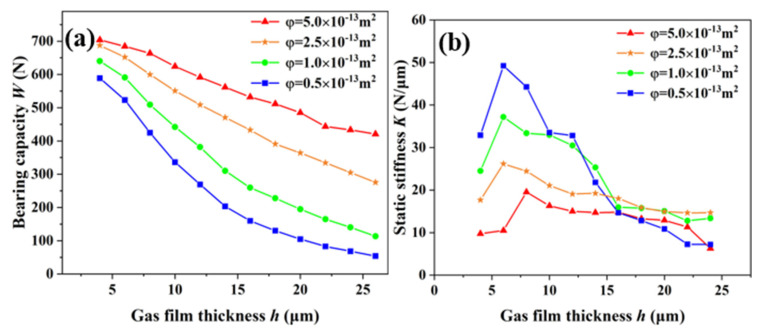
Static characteristics of aerostatic bearing under different permeabilities: (**a**) bearing capacity; (**b**) static stiffness.

**Figure 9 membranes-12-00898-f009:**
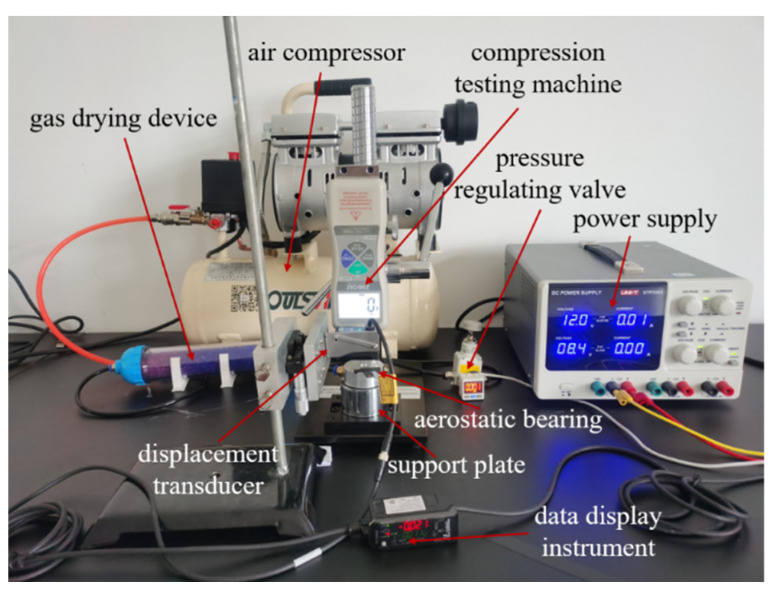
Experimental device for measuring static characteristics of aerostatic bearing.

**Figure 10 membranes-12-00898-f010:**
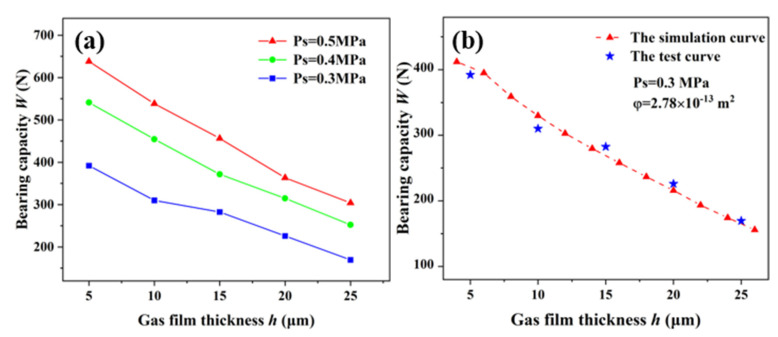
Bearing capacities of aerostatic bearing: (**a**) experimental results; (**b**) comparison of theoretical calculation and experimental results.

**Table 1 membranes-12-00898-t001:** The flow measurement data of porous SiC ceramic.

	1	2	3	4
Pressure drop (Δ*p*) (MPa)	0.02	0.03	0.04	0.05
Volume flow rate (*Q ×* 10^−5^ (m^3^/s))	8.633	13.283	18.583	23.267

**Table 2 membranes-12-00898-t002:** The parameters of the aerostatic bearing.

Parameter	Value
Bearing body diameter (*D*_0_) (mm)	48
Gas-capacity thickness (*H*_0_) (mm)	10
Porous medium diameter (*D*_1_) (mm)	43
Porous medium thickness (*H*_1_) (mm)	5
Gas-film thickness (*h*) (μm)	4~26
Gas-supply pressure (*P*_s_) (MPa)	0.5
Environmental temperature (*T*) (K)	298
Environmental pressure (*P*_0_) (MPa)	0.1
Aerodynamic viscosity (*μ*) (N·s/m^2^)	1.7894 × 10^−5^
Permeability coefficient of porous medium (*φ*) (m^2^)	2.78 × 10^−13^

## Data Availability

Not applicable.

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
