# Peer review of "Study on Static Characteristics of Aerostatic Bearing Based on Porous SiC Ceramic Membranes"

_membranes, 2022, doi:10.3390/membranes12090898_

Round 1
Reviewer 1 Report
The work by Xiao et al on aerostatic bearing based on porous SiC ceramic membrane depicts experimental analysis and theoretical verification on the same. The work is interesting with simple experimental design which brings out the essence of the work. The work can be accepted with following changes as advised-
1. Line 79-80: Authors should elaborate the SIC porous membrane fabrication detail, like- the constituents' ratio, the applied vacuum etc.
2. Authors should provide the details on porosity measurements, since it's the single most important aspect. Just citing two references is a bit sloppy.
3. Figure 3: It seems the value of permeability is unneeded to be shown in figure. Eventually, it would come as a constant value from simple Y=mx relation between Q and dP. Now the LHS Y axis is confusing.
4. Authors should provide the simulation parameters, with mesh count, steady state parameters and grid independence/sensitivity measures.
5. Figure 10b: The applied pressure should be written in the figure description.
6. It is understood that authors have used standard theoretical models. However, their USP is the experimental simplicity. In this case authors should compare their data with other researchers.
7. Authors should also provide the details of mechanical properties of the SIC membrane.
Reviewer 2 Report
The present manuscript is more of a technical manuscript than a scientific one. Basically it is the use of a finite element program to solve a specific problem.
The Darcy's law that they use has many restrictions that must be fulfilled, so they must be verified, such as the Reynolds number and the capillary number. An exhaustive work of all conditions has been extensively studied by Professor Narciso et al. [1-3]. A very didactic approach can also be found in the following book chapter [4].
Regarding sintering. They must be much more precise in the data, you cannot say they are mixed in a certain amount, you have to give the proportions of all the components, the used particle size of all the components, and that of the SiC the particle size distribution . sintering atmosphere. It is not the same in air as in Ar, for example, the way of sintering is totally different. And based on that explain what is happening. Why has 145º ºC and 2 hours been chosen, and what is the heating ramp, etc….
An Hg porosimetry should be done to determine the type of pore with greater precision, it is not enough to say the porosity is 25%.
At least 3 membranes should be presented, for example keeping everything constant and changing the sintering time, or doing 3 temperatures (1400, 1450, 1500 ºC) see the following article will be of great help [5].
It is not necessary to write all the equations, they are general and can be obtained in many fluid mechanics books, or even in the manual of the finite element program. Have you tried to change the mesh to see the convergence and precision?
In conclusion, the manuscript cannot be published in the present form, it still needs a lot of work and to be more rigorous and precise.
Round 2
Reviewer 1 Report
The article can be accepted once the authors show the grid independence study with convergence level up to 10-6 with near flat lines. Now the results are steady declining and they haven't of course converged. Also, the authors must discuss about the number of nodes taken.
Reviewer 2 Report
Dear authors,
I am pleased to see that you have greatly improved your manuscript, however you have ignored all my recommendations about references, which are essential in your manuscript.
Point 1. I stand by my comments.
Point 2. It seems correct to me. Just add that in reality you are not doing sintering if not what you are doing is reactive infiltration [5,6], the silicon reacts with the carbon that has been formed in the preform during the pyrolysis process, so you must modify that paragraph.
Point 3. It seems correct to me. However something strange is happening, if you have particles of 20 micrometers, the pores cannot be 20 micrometers. Much of the research in the first 4 references deals with this topic. Keep in mind that this equation is intended for cylinders, also to think that the contact angle of Hg is universal is incorrect, although most researchers accept it... it should be commented.
Point 4. I could accept it, depending on the modification and improvement of the first 3 points.
Point 5. It seems correct to me.
Therefore, my recommendation is that it could be accepted if all the aforementioned modifications are made, including the inclusion of the references
1. doi.org/10.1016/S1359-6454(99)00318-3
2. doi.org/10.1016/S1359-6454(01)00348-2
3. 10.1016/j.cossms.2006.02.007
4. library.oapen.org/handle/20.500.12657/49131
5. doi.org/10.3390/ma12152425
6. doi.org/10.1111/j.1551-2916.2009.03572.x
Round 3
Reviewer 1 Report
The article can be accepted.
Reviewer 2 Report
Dear authors,
I think that at this moment everything is clear, both the experimental part and everything related to finite elements. It would have been a plus if you had made more membranes with different porosity and pore size. Now it's good work, but it could have been very good.
My recommendation is that the manuscript be published, however I have found a serious error in a reference [reference 26]. So they should give one last review, from the quiet, to polish the English, and check that there are no more errors.
